# Effects of Thermal Stress on the Antioxidant Capacity, Blood Biochemistry, Intestinal Microbiota and Metabolomic Responses of *Luciobarbus capito*

**DOI:** 10.3390/antiox12010198

**Published:** 2023-01-14

**Authors:** Kun Guo, Rui Zhang, Liang Luo, Shihui Wang, Wei Xu, Zhigang Zhao

**Affiliations:** 1Key Open Laboratory of Cold Water Fish Germplasm Resources and Breeding of Heilongjiang Province, Heilongjiang River Fisheries Research Institute, Chinese Academy of Fishery Sciences, Harbin 150070, China; 2Engineering Technology Research Center of Saline-Alkaline Water Fisheries (Harbin), Chinese Academy of Fishery Sciences, Harbin 150070, China

**Keywords:** high temperature, biochemistry, intestinal microbial, metabolome, *Luciobarbus capito*

## Abstract

The rise in water temperature caused by global warming is seriously threatening the development of aquatic animals. However, the physiological response mechanism behind the adverse effects of thermal conditions on *L. capito* remains unclear. In this study, we investigated the physiological responses of *L. capito* exposed to thermal stress via biochemical analyses and intestinal microbiota and liver LC–MS metabolomics. The results show that the superoxide dismutase (SOD) and catalase (CAT) activities significantly decrease, while the malondialdehyde (MDA) content, aspartate aminotransferase (AST), acid phosphatase (ACP), alanine aminotransferase (ALT), and albumin (ALB) activities, and glucose (Glu) level significantly increase. Obvious variations in the intestinal microbiota were observed after stress exposure, with increased levels of Proteobacteria and Bacteroidota and decreased levels of Firmicutes, Fusobacteriota, and Actinobacteriota, while levels of several genera of pathogenic bacteria increased. Liver metabolomic analysis showed that stress exposure disturbed metabolic processes, especially of amino acids and lipids. The results of this study indicated that thermal stress caused oxidative stress, disturbed blood biological functioning and intestinal microbiota balance, and damaged amino acids and lipids metabolism of liver in *L. capito*.

## 1. Introduction

With the increase in human activities, the degree of global warming has been aggravated in recent decades, causing water temperatures to rise, which has had a negative impact on the aquaculture industry [1]. Water temperature is a critical ecological factor affecting the physiological and biochemical state of aquatic ectotherms [2,3,4]. Although a small range of temperature changes is usually beneficial to fish [5,6], changes that exceed a certain range will disrupt the normal physiological balance of fish [7]. Several studies have indicated that a sudden rise in water temperature disrupts metabolic characteristics, weakens immune defense, and increases the incidence of pathogenic bacteria development in fish [8,9,10].

A sharp increase in ambient temperature is defined as heat shock, thermal shock, or high-temperature shock, and can cause a rapid increase in fish body temperature, resulting in strong behavioral and physiological responses [11,12,13]. Thermal stress can result in the generation of large amounts of reactive oxygen species (ROS), including hydroxyl radicals (·OH), superoxide (O^2−^), and hydrogen peroxide (H_2_O_2_) [14,15]. An excessive amount of ROS can destroy the oxidation defense system and thus cause oxidative stress [16]. Toxic ROS accumulate in fish, destroying many cellular components, and leading to oxidative damage, lipid peroxidation, protein damage, DNA expression change, enzyme inactivation, and other phenomena [17]. In the antioxidant system of fish, enzyme antioxidants including superoxide dismutase (SOD), catalase (CAT), glutathione-S transferase (GST), etc., scavenge ROS through the enzymatic system when the body is damaged by oxidative stress. SOD is considered to play a key role in the first step of the enzymatic antioxidative defense system, catalyzing the dismutation of superoxide radicals into H_2_O_2_ and O_2_. The end product of the dismutation reaction, H_2_O_2_, can be removed by CAT [18]. As the final product of lipid peroxidation caused by ROS, malondialdehyde (MDA) reflects the level of lipid peroxidation [19]. Therefore, these indicators can be used as key oxidative stress biomarkers.

Blood plays an important role in the physiological state and immune defense of fish [20], and environmental stress factors can change the values of the blood index [21,22]. Therefore, blood biochemical parameters are often used to assess physiological health under stressful conditions [23]. The liver is the main organ responsible for metabolism in animals [24], and it plays a core role in the absorption and metabolism of nutrients, and elimination of toxic substances [25]. Therefore, the liver is usually selected as a target organ for studying biological processes in response to biotic and abiotic factors. The intestine is an important organ that plays a vital role in maintaining the host healthy biological processes in the host [26]. The intestines are rich in microbial communities, which participates in many important physiological processes, such as the digestion and absorption of nutrients, osmotic pressure regulation, and immune functions [27,28], and are sensitive to changes in the environment [29]. Metabolomics is an efficient and accurate technique for analyzing most small molecule metabolites in biological samples and has been used widely in fish studies, including for the monitoring of fish nutritional status and other changes in life activities [30,31]. Metabolomics is also used for studying the impacts of environmental changes on the physiological metabolism of aquatic animals and can provide deep insights into how aquatic animals respond to natural stressors [32,33].

*Luciobarbus capito* belongs to the family Cyprinidae and subfamily Barbinae, to which the closely related genus *Barbus* also belongs, and it is native to the Aral Sea [34]. In 2003, the species was first introduced to China by the Heilongjiang River Fisheries Research Institute, Chinese Academy of Fishery Sciences, and its proliferation was subsequently promoted in many cities. *L. capito* has the characteristics of delicious taste, rapid growth, and strong resistance to adversity, and has become an economically important fish in China [35]. Evidence indicates that the range of water temperatures suitable for *L. capito* is 18–27 °C, with an optimal temperature range of 24–27 °C [36]. The average temperatures gradually rise, with the maximum water temperatures of aquaculture ponds in summer often reaching 35 °C in Southern China in recent years. Thus, this species may experience thermal stress. However, research on the physiological and molecular mechanisms of heat stress response is scarce, and tends not to be systematic. Here, we performed enzyme activity studies related to antioxidant and nonspecific immunity, intestinal microbiota, and untargeted LC–MS metabolomics analysis, focusing on the response of *L. capito* to thermal stress. The study preliminarily reveals the physiological and molecular mechanisms of the heat stress response in *L. capito* and provides a theoretical basis for further research on its healthy breeding technology.

## 2. Materials and Methods

### 2.1. Ethics Statement

All animal procedures in this study were conducted according to the guidelines for the care and use of laboratory animals of Heilongjiang River Fisheries Research Institute, CAFS. The studies in animals were reviewed and approved by the Committee for the Welfare and Ethics of Laboratory Animals of Heilongjiang River Fisheries Research Institute, CAFS. The project identification code is 20220420-001, the date of approval was 30 April 2022, and the name of the ethics committee is the Committee for the Welfare and Ethics of Laboratory Animals of Heilongjiang River Fisheries Research Institute, CAFS. This study complies with all relevant laws.

### 2.2. Thermal Stress

The *L. capito* specimens, with a mean body weight of (30.85 ± 2.62) g, were collected from the Hulan experimental station of the Heilongjiang Fishery Research Institute, Chinese Academy of Fishery Sciences in Hulan District, Harbin City, Heilongjiang Province, China. The fish were transported to the laboratory and acclimatized in rearing tanks containing aerated water (water temperature 22 ± 1 °C; dissolved oxygen > 6.5 mg/L; pH 7.8 ± 0.2) for one week prior to the experiment. The size of the glass tank was 87.5 cm × 52.5 cm × 50.0 cm, and it contained 200 L of water. The fish were fed with commercial formula feed at 8:00 and 17:00 every day until 24 h before the experimental treatments.

### 2.3. Experimental Design and Sampling

After acclimation, *L. capito* were randomly divided into two groups: the control group (CG) and the thermal stress group (HT). Each group included six replicate tanks, with 20 individuals per tank. Two automatic temperature control systems were used for water temperature control. In the HT group, the water temperature of the six tanks increased from 22 to 33 °C at a rate of 1 °C/h. In the CG group, the water temperature of the six tanks was kept at 22 °C. The stress exposure lasted for 96 h.

After 96 h of exposure, the intestines, blood, and liver of numerous *L. capito* fish were sampled from different tanks and then were mixed according to the same treatment for analysis. Each treatment group contained 3 biologically repeated biochemical samples, 5 biologically repeated microbial samples, and 6 biologically repeated metabolome samples.

### 2.4. Biochemical Analysis

After the blood was first stored in the dark at 4 °C for 4 h, it was centrifuged for 10 min to obtain serum. A total of 0.20 g of liver tissue was first weighed and then homogenized in 9 vol (*w*/*v*) of isotonic physiological saline. The homogenate was then centrifuged at 2500 rpm, 4 °C, for 10 min, followed by collection of the supernatant for later use. The activities of superoxide dismutase (SOD) and catalase (CAT) enzymes and the malondialdehyde (MDA) content of liver were measured using specific commercial kits (Nanjing Jiancheng Bioengineering Institute, Nanjing, China). The aspartate aminotransferase (AST), acid phosphatase (ACP), alanine aminotransferase (ALT), and albumin (ALB) activities and the glucose (Glu) levels of serum were detected using a Mindray BS-240VET Automated Hematology Analyzer (Mindray Bio-Medical Electronics Equipment, Co., Ltd., Shenzhen, China). Statistical analyses were performed using SPSS 19.0. Differences between thermal exposure stress and control treatments were determined by *t*-test and differences were considered significant at *p* < 0.05.

### 2.5. Intestinal Microbiome Analysis

Total genomic DNA of the intestinal microbes was extracted using a DNeasy^®^ PowerSoil^®^ Pro Kit (QIAGEN, Hilden, Germany) according to the manufacturer’s instructions. The DNA extract was checked on 1% agarose gel, and the DNA concentration and purity were determined using a NanoDrop 2000 UV–vis spectrophotometer (Thermo Scientific, Wilmington, MA, USA). The hypervariable V3–V4 region of the bacterial 16S rRNA gene was amplified with the primer pair 338F (5′-ACTCCTACGGGAGGCAGCAG-3′) and 806R (5′-GGACTACHVGGGTWTCTA AT-3′) on an ABI GeneAmp^®^ 9700 PCR thermocycler (ABI, San Diego, CA, USA). The PCR amplification of the 16S rRNA gene was performed as follows: initial denaturation at 95 °C for 3 min followed by 27 cycles of denaturing at 95 °C for 30 s, annealing at 55 °C for 30 s, and extension at 72 °C for 45 s, and a single extension at 72 °C for 10 min, with holding at 4 °C. The PCR mixtures contained 5× TransStart FastPfu buffer 4 μL, 2.5 mM dNTPs 2 μL, forward primer (5 μM) 0.8 μL, reverse primer (5 μM) 0.8 μL, TransStart FastPfu DNA Polymerase 0.4 μL, template DNA 10 ng, and finally ddH2O, to a final reaction volume of 20 μL. PCRs were performed in triplicate. The PCR product was extracted from 2% agarose gel and purified using the AxyPrep DNA Gel Extraction Kit (Axygen Biosciences, Union City, CA, USA), according to the manufacturer’s instructions, then quantified using a Quantus™ Fluorometer (Promega, Madison, WI, USA). Purified amplicons were pooled in equimolar amounts and paired-end sequenced on an Illumina MiSeq PE300 platform (Illumina, San Diego, CA, USA).

Raw FASTQ files were de-multiplexed using an in-house Perl script and then quality-filtered using fastp version 0.19.6 and merged using FLASH version 1.2.7. Then, the optimized sequences were clustered into operational taxonomic units (OTUs) using UPARSE 7.1 with a 97% sequence similarity level. Based on the OTU information, rarefaction curves and alpha diversity indices, including observed OTUs, Chao1 richness, and Shannon index, were calculated using Mothur v1.30.1. A Venn diagram was used to count the number of unique and shared OTUs in different groups. The similarity among the microbial communities in different samples was determined using principal coordinate analysis (PCoA) based on Bray–Curtis dissimilarity using the Vegan v2.5-3 package. Microbial composition was analyzed at the phylum, class, and genus levels.

### 2.6. Liver Metabolomics Analysis

The liver samples from two groups were obtained for the metabolomics study using the UHPLC–MS/MS platform (Thermo, Ultimate 3000LC, Q Exactive, Wilmington, MA, USA). Frozen liver samples were thawed at 4 °C, and 50 mg of sample was accurately weighed and transferred into a 1.5 mL microcentrifuge tube. Then, 400 µL of extraction solution (methanol/water = 4:1) was added to each sample. The mixture was homogenized using a homogenizer (Wonbio-96c, Shanghai, China) at 50 Hz for 6 min followed by vortexing for 30 s and ultrasound at 40 kHz for 30 min at 5 °C. The samples were placed at −20 °C for 30 min to precipitate proteins. After centrifugation at 13,000× *g* at 4 °C for 15 min, the supernatants were carefully transferred to sample vials for LC–MS/MS analysis. The quality control (QC) samples were prepared by mixing aliquots to obtain a pooled sample and then analyzed accordingly. Samples were injected at regular intervals (every 6 samples) to monitor the stability of the analysis. A multivariate statistical analysis was performed using R package. Principle component analysis (PCA) using an unsupervised method was applied to obtain an overview of the metabolic data, and general clustering, trends, and outliers were visualized. All of the metabolite variables were scaled to unit variances prior to conducting the PCA. Orthogonal partial least squares discriminate analysis (OPLS-DA) was used for statistical analysis to determine global metabolic changes between comparable groups. All of the metabolite variables were scaled to Pareto scaling prior to conducting the OPLS-DA. The model validity was evaluated from model parameters R^2^ and Q^2^, which provide information for the interpretability and predictability, respectively, of the model and avoid the risk of over-fitting. Variable importance in the projection (VIP) was calculated using the OPLS-DA model.

### 2.7. Correlation Analysis of Intestinal Bacteria and the SDEMs

Pearson correlation analysis was employed to reveal the correlation between intestinal bacteria and liver SDEMs using the Cytoscape software. *p* < 0.05 was regarded as a statistically significant difference, *p* < 0.01 was regarded as very significant, and *p* < 0.001 was regarded as extremely significant. A heatmap was used to illustrate the correlations of intestinal bacteria metabolites.

## 3. Results

### 3.1. Biochemical Parameter Changes

After thermal stress, the activities of SOD and CAT were significantly lower in the HT group than in the CG group (Figure 1A,B). Conversely, the MDA contents, AST, ALT, ACP, and ALB activities, and Glu levels were significantly higher in the HT group than in the CG group (Figure 1C–H).

### 3.2. Intestinal Microbiota Changes

#### 3.2.1. Richness and Diversity

A total of 337,024 effective sequences were obtained from 10 samples with an average length of 427 bp per sample. The rarefaction curve tended to reach a plateau (Figure 2A), which indicates that there was adequate sequencing depth in all the samples, and the data can thus be used for subsequent analysis in this study. A total of 219 shared OTUs were detected in the two groups, and the number of unique OTUs for the CG and HT groups was 86 and 136, respectively (Figure 2B). Compared with the CG group, the Shannon index value of the HT group increased, while the values of the Simpson, Ace, and Chao indices decreased, but these differences were not statistically significant (*p* > 0.05) (Figure 2C). The PCoA analysis based on Bray–Curtis showed that the intestinal bacteria in the CG and HT groups were clearly separated (Figure 2D).

#### 3.2.2. Intestinal Microbial Composition

The taxa of dominant bacteria among the two groups were similar, while their abundance was altered. At the phylum level, the relative abundance of Proteobacteria and Bacteroidota increased in the HT group compared with the CG group, while the abundance of Firmicutes, Fusobacteriota, and Actinobacteriota species decreased (Figure 3A). At the class level, the relative abundance of Gammaproteobacteria, Bacteroidia, and Clostridia increased, while the abundance of Bacilli, Fusobacteriia, and Verrucomicrobiae decreased (Figure 3B). At the genus level, certain genera, such as *Acinetobacter*, *Chryseobacterium*, and *Pseudomonas*, increased in the HT group, while *Citrobacter*, *Cetobacterium*, and *Bacteroides* decreased (Figure 3C).

### 3.3. Liver Metabolomics Analysis

The UHPLC–MS/MS platform was used to perform untargeted metabolomics analysis of liver samples to investigate the metabolic changes in *L. capito*.

The unsupervised PCA score plots show that the metabolic profiles of the thermal stress group changed significantly when compared with the control group in both positive and negative ion modes (Figure 4A,B). The OPLS-DA model (model evaluation parameters: positive ion mode: R^2^Y = 0.961 cum, Q^2^ = 0.949 cum; negative ion mode: R^2^Y = 0.977 cum; Q^2^ = 0.964 cum) was established for both negative and positive ion modes, and the model is shown to be stable and reliable (Figure 4C,E). Next, we used the permutation test to establish 200 OPLS-DA models in which the order of the categorical variables Y was randomly changed to obtain the R^2^ and Q^2^ values of the stochastic model (Figure 4D,F). In the figures, from left to right, all Q^2^ points are lower than the original blue Q^2^ points on the right, which indicates that the model is robust and reliable with no overfitting. The data above indicate that the obtained model had a good fitting ability and high predictability and is suitable for subsequent DEM analysis.

VIP value > 1.0 and *p* < 0.05 were used as standards to select the DEMs. We identified 232 DEMs, comprising 140 upregulated and 92 downregulated metabolites. Crossed heatmaps (Figure 5A) show the DEMs between the two groups.

Based on multivariant analysis, the significantly DEMs between the HT and CG groups were identified, and parameters such as the compound name, VIP value, *p*-value, fold change, and variation in these metabolites are presented in Table 1.

To explore the potential metabolic pathways affected by thermal stress, the KEGG annotations of all DEMs were further analyzed. Compared with CG, the most enriched pathways in the HT group based on the DEMs were alpha-linolenic acid metabolism, pantothenate and CoA biosynthesis, isoflavonoid biosynthesis, glycerophospholipid metabolism, ascorbate and aldarate metabolism, caffeine metabolism, and arginine biosynthesis (Figure 5B).

### 3.4. Association between the Intestinal Microbiota and Metabolites

To reveal the relationships between intestinal microbial and metabolite parameters, heatmaps were generated via Pearson correlation analysis. The abundance of several of the bacterial genera discussed above was significantly correlated with the levels of metabolites (Figure 6). For example, *Acinetobacter* abundance was positively correlated with changes in gluconic acid, L-histidine, and L-lysine but negatively correlated with changes in PC(22:5(4Z,7Z,10Z,13Z,16Z)/22:6(4Z,7Z,10Z,13Z,16Z,19Z)) and LysoPC(22:4(7Z,10Z,13Z,16 Z)).

## 4. Discussion

### 4.1. Antioxidant and Blood Biochemistry Responses to Thermal Stress

Oxidative stress is a significant toxic mechanism of environmental stress that impacts aquatic animals [37,38]. Previous studies have demonstrated that adverse environmental conditions induce ROS production, and excessive ROS might attack DNA, proteins, lipids, and other cellular biological macromolecules [17]. Lipid peroxidation is considered to be a consequence of the oxidation of lipids by ROS [39]. MDA is the product of lipid peroxidation, and its levels can indirectly reflect the degree of oxidative stress [19]. In this experiment, the MDA content increased significantly in the liver of *L. capito* after thermal stress, indicating the accumulation of lipid peroxidation in the liver.

Fortunately, aquatic organisms possess an antioxidant defense system that mitigates the deleterious effects of ROS [40]. Under normal physiological conditions, the enzyme system and antioxidants in the body can maintain a balance between ROS production and clearance. When the antioxidant capacity is unbalanced and severely biased toward ROS generation, this can lead to an increase in oxygen ions and free radical peroxides, which can effectively induce oxidative stress [41]. The antioxidant enzymes are the first line of intracellular defense against oxidative stress [16]. Our results showed that the activities of SOD and CAT in the *L. capito* liver were significantly decreased under thermal stress. These results indicate that heat stress weakened antioxidant defense in the body, disturbed physiological homeostasis, and led to oxidative damage.

Serum biochemical indexes reflect the metabolic status of the body. When fish experience adversity (hunger, temperature fluctuation, crowding, etc.), they initiate their own resistance mechanism, thus regulating their respiration and energy metabolism [42,43]. Glu is one of the major sources of energy, and its levels can directly reflect changes in the energy metabolism in fish and have a certain correlation with the stress response of fish, so it is widely used to monitor the metabolic changes in fish during stress [44]. The changes in glucose are known to be a secondary stress response under stress environments. It has been found that after entering a stress environment, fish will use glycogen decomposition and glycosylation to generate the energy needed to resist stress pressure, thus reducing the amount of glycogen in the body [45]. Fish can maintain their blood sugar balance through glucose metabolism, thus appropriately adjusting their glucose levels. For example, the blood Glu level of *Scophthalmus maximus* increases significantly under low-temperature stress [46]. In this study, the serum Glu level was found to be significantly higher in the thermal stress group than in the control group. This finding implies that *L. capito* consumed more energy to cope the stressful situations as in thermal stress. ALT and AST are two important transaminases that can reflect the metabolic state and material transformation as well as changes in tissue structure and function [47]. Their activities are important indicators of cell membrane integrity [42]. Under normal physiological metabolic conditions, ALT and AST mainly exist in the liver, while serum ALT and AST levels are low and relatively stable. However, when the liver is damaged due to stress, the permeability of the tissue cell membrane increases, and the ALT and AST in the tissue enters the blood through the cell membrane, causing increased activity of ALT and AST in the serum [48]. In this study, the serum ALT and AST levels in the thermal stress group were found to be significantly higher than those in the control group, indicating that high-temperature stress had caused tissue damage in *L. capito*. ALB plays an important role in maintaining the stability of blood osmotic pressure, participating in the processes of nutrition transportation, coagulation, anticoagulation, hepatocyte repair, and regeneration to maintain a stable chemical environment in the body [49]. The results of this study show that the serum ALB content of fish significantly increased after high-temperature stress. The reason may be that as the carrier of nutrients, ALB increases in content in response to high temperature stress, thus providing energy for the body and participating in maintaining the balance of plasma colloid osmotic pressure. With increasing exposure time, the fish were stimulated by the high temperature and the liver was damaged to a certain extent, which affected the protein metabolism of the body and led to an increase in serum ALB content. ACP is an important part of the innate immune system, and also participates in phosphorylation and dephosphorylation processes in fish, playing a crucial role in metabolism [50]. In this study, high-temperature stress increased the levels of ACP, indicating that the innate immune system of the fish was activated to resist the adverse environment.

### 4.2. Intestinal Microbiota in Response to Thermal Stress

Environmental factor stress has negative effects on the intestinal immune barrier function of aquatic animals [51]. Research has shown that high temperatures will directly damage the intestinal mucosa, affect intestinal flora colonization, and lead to potential opportunistic pathogens in the intestine infiltrating other organs, endangering the health of the host [52]. For example, hypoxia stress was found to significantly change the intestinal microflora structure of *Rachycentron canadum* [53].

In this study, high-throughput sequencing was used to study the effects of thermal stress on the intestinal microflora of *L. capito*. The experimental results showed that Proteobacteria, Firmicutes, Actinobacteria, and Bacteroidota are the dominant intestinal bacteria of *L. capito*, which is consistent with the results of previous research [54].

Temperature can directly alter the gut microbiome in ectotherms and the growth of microorganisms. Previous studies have shown that the damage of temperature to the intestinal microbiota of animals may reduce the host’s resistance to pathogenic microorganisms [55]. The results of PCoA diversity analysis revealed that thermal stress altered the structure of the intestinal microflora in *L. capito*. The dominant microflora at the phylum level in the thermal and control groups did not change, showing that temperature did not affect the occurrence of the dominant microflora but changed their relative abundance. Proteobacteria participate in carbon complexes and nitrogen degradation, and their increase may lead to an imbalance in the intestinal flora and an increased risk of enteritis [56,57]. Firmicutes and Bacteroidota play a vital role in the energy metabolism and lipid metabolism of the host [58]. In this study, the relative abundances of Proteobacteria and Bacteroidota increased, while the abundance of Firmicutes decreased, which suggests that thermal stress threatens the host’s health status. At the genus level, several dominant genera exhibited obvious differences between the two groups. The intestines of fish contain some conditional pathogenic bacteria, such as Acinetobacter and Pseudomonas, which are commonly present in water and are recognized as important agents of animal diseases [59]. This may be caused by the fact that high temperature is more conducive to the growth of these pathogenic bacteria. Therefore, the abundance of pathogenic bacteria may increase the risk of disease development in *L. capito*.

### 4.3. Liver Metabolomics Analysis in Response to Thermal Stress

Liver metabolomics analysis further demonstrated that thermal stress affects the metabolic function of *L. capito*. Amino acids are biologically important molecules that can also serve as an energy source, and they are important in biological processes such as growth, reproduction, and immunity [60,61]. The contents of five amino acids (L-lysine, L-histidine, L-arginine, argininosuccinic acid, and L-tryptophan) were changed significantly in thermally challenged fish compared with control fish. A pathway related to amino acid metabolism (arginine biosynthesis) was enriched. Lysine is a limiting amino acid that participates in protein synthesis [62]. Arginine is the main substrate for NO production, and it plays an important role in nitrogen metabolism, creatine and polyamine synthesis. In addition, arginine is also involved in tissue repair, cell replication and immune regulation in fish [63]. Histidine is a precursor molecule for neurohormones and neurotransmitters that helps to regulate the balance of the histaminergic system [64]. Histidine is also a glycogenic amino acid, which can transform glucose and glycogen to provide energy for the organism. Under adverse environments, fish need more energy to resist stress, while the conversion of amino acids into glucose as an energy source is an important response strategy [65]. Under thermal stress, the content of L-histidine decreased, indicating that L-histidine may be used as an energy substrate, while the content of lysine increased, indicating that it may be used as functional substrate. The levels of L-histidine and L-lysine in response to thermal stress were significantly changed in the liver, and the arginine biosynthesis metabolic pathway was enriched, which suggests that amino acid metabolism was induced by thermal stress. Lipid metabolism participates in fatty acid and energy storage and environmental stress responses in aquatic animals [66]. The results suggested the role of lipid metabolism in supporting the thermal response of *L. capito*. In this work, metabolomics data show that thermal stress results in significantly changes to lipid homeostasis in the fish liver. The metabolic pathways related to lipid metabolism, alpha-linolenic acid metabolism, and glycerophospholipid metabolism were significantly altered. Levels of seven lipids (e.g., LysoPC(20:2(11Z,14Z)) and LysoPC(16:0)) changed significantly in thermally challenged fish compared with control fish. Phosphoglycerides include lecithin (PC), cephalin (PE), and phosphatidylserine (PS), which are important components of the cell membrane [67]. Lysophospholipids (LysoPC) participate in biological processes in most cells, affecting the immune system by regulating immune cells [56]. The increase in PCs plays an important role in animals’ resistance to heat stress [68]. Under thermal stress, the levels of PCs (PC(22:6(4Z,7Z,10Z,13Z,16Z,19Z)/22:6(4Z,7Z,10Z,13Z,16Z,19Z)) and PC(22:5(4Z,7Z,10Z,13Z,16Z)/22:6(4Z,7Z,10Z,13Z,16Z,19Z))) and LysoPCs (LysoPC(20:2(11Z,14Z)), LysoPC(16:0), LysoPC(22:4(7Z,10Z,13Z,16Z)), and LysoPC(22:6(4Z,7Z,10Z,13Z,16Z,19Z))) decreased, which showed that 96h high-temperature stress had exceeded the tolerance of *L. capito*, breaking the balance of lipid metabolism. Unsaturated fatty acids are not only important components of the cell membrane, but also participate in energy metabolism. In this study, the content of unsaturated fatty acids with important functions decreased. Collectively, based on the above results, we speculate that thermal exposure causes lipid metabolism disorder and influences the fish liver by disrupting the structure and function of the cell membrane and mitochondria.

### 4.4. Correlation between Intestinal Bacteria and Metabolism

Changes in intestinal microbiota can affect the metabolism of the host [20]. In this study, the abundance of several of the bacterial genera discussed above was significant correlated with the levels of certain metabolites. For example, Acinetobacter abundance was positively correlated with changes in gluconic acid, L-histidine, and L-lysine but negatively correlated with changes in PC and LysoPC. The identification of sensitive biomarkers is one of the aims of aquatic toxicology. In this study, several sensitive metabolite markers were identified, which will be helpful in exploring the toxicity mechanisms of thermal stress.

## 5. Conclusions

By applying traditional biochemical approaches, LC–MS-based untargeted metabolomics, and microbiome analysis, the results of our study provide novel insights into the molecular mechanisms underlying the liver response of *L. capito* to acute thermal stress. Thermal stress was found to cause oxidative stress and disturb the immune system of *L. capito*. Thermal stress also caused variations in the intestinal bacterial community and increased the abundance of harmful bacteria. At the same time, thermal stress disrupted amino acid metabolism and lipid metabolism. The abundances of several intestinal bacterial genera were shown to be correlated with the metabolic functioning of their hosts.

## Figures and Tables

**Figure 1 antioxidants-12-00198-f001:**
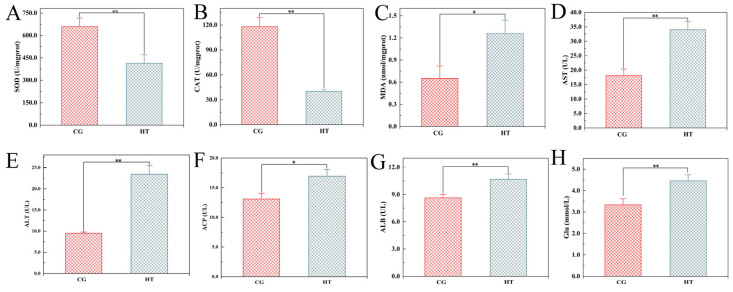
Changes in biochemical indicators in *L. capito* after thermal stress. Bars indicate mean ± SD (n = 3). (**A**) Superoxide dismutase (SOD), (**B**) catalase (CAT), and (**C**) malondialdehyde (MDA) in liver; (**D**) aspartate aminotransferase (AST), (**E**) alanine aminotransferase (ALT), (**F**) acid phosphatase (ACP), (**G**) albumin (ALB), and (**H**) glucose (Glu) in serum. * represents significant difference (*p* < 0.05), and ** represents highly significant difference (*p* < 0.01).

**Figure 2 antioxidants-12-00198-f002:**
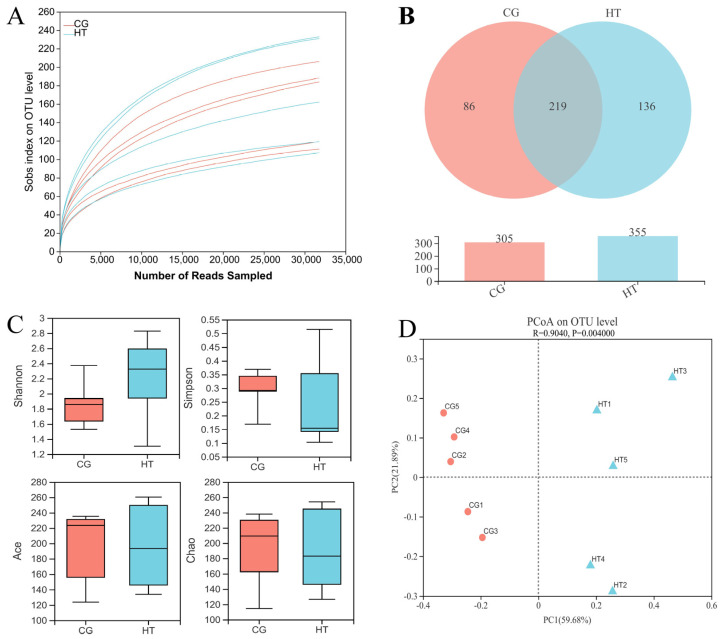
The richness and diversity of intestinal microbial communities in *L. capito* after thermal stress. (**A**) Rarefaction curve. (**B**) Venn diagram. (**C**) Alpha diversity indices. Bars represent the mean ± SD (n = 5). (**D**) Beta diversity indicated by PCoA based on Bray–Curtis distance.

**Figure 3 antioxidants-12-00198-f003:**
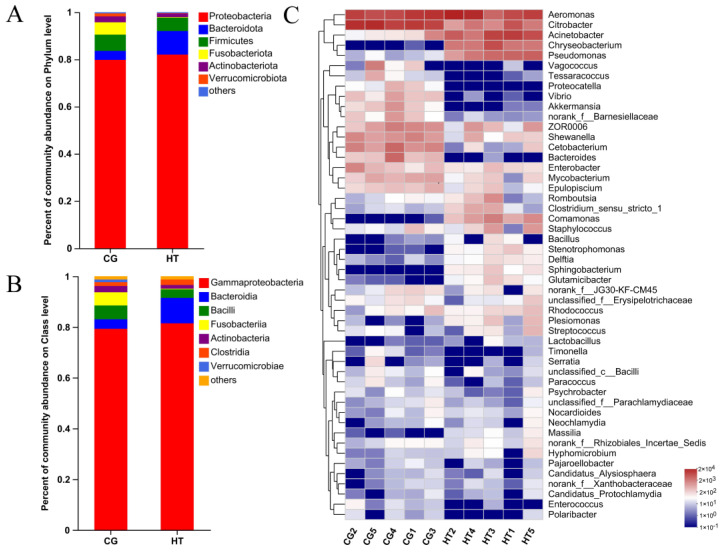
Changes in intestinal microbial composition of *L. capito* after thermal stress. Relative microbial abundance at the (**A**) phylum and (**B**) class level. (**C**) Heatmap analysis of the top 50 dominant bacterial genera. Red color indicates higher abundance of the genera, and blue color indicates lower abundance, with the intensity of color reflecting the degree of change as indicated in the scale.

**Figure 4 antioxidants-12-00198-f004:**
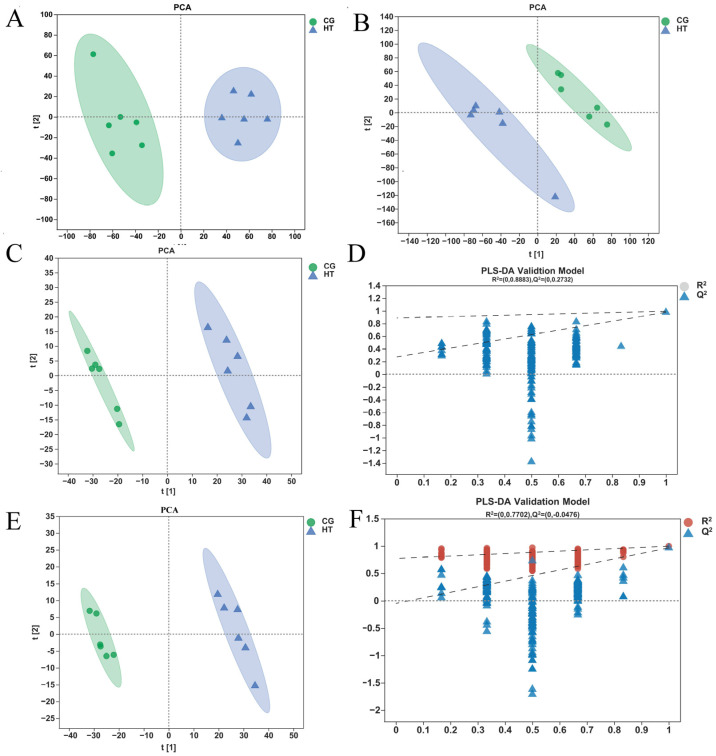
Quality analysis of metabolomics data. The PCA score plot of samples acquired in (**A**) positive ion and (**B**) negative ion mode. The OPLS-DA (**C**) score plot and (**D**) permutation test for positive ion mode. The OPLS-DA (**E**) score plot and (**F**) permutation test for negative ion mode.

**Figure 5 antioxidants-12-00198-f005:**
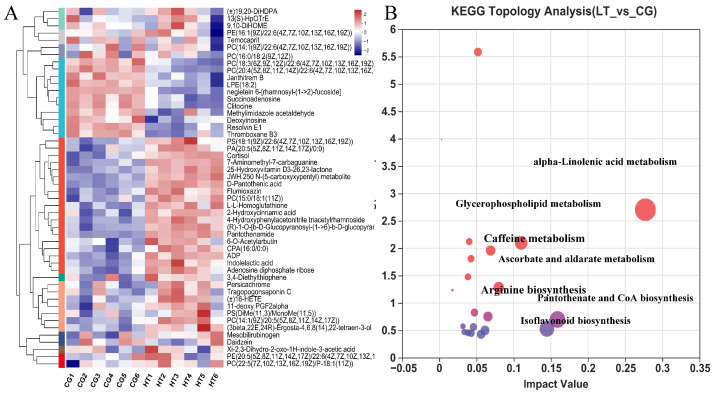
(**A**) Hierarchical clustering and analysis of the (**B**) most enriched KEGG pathway of DEMs in the hemolymph of *L. capito* after thermal stress.

**Figure 6 antioxidants-12-00198-f006:**
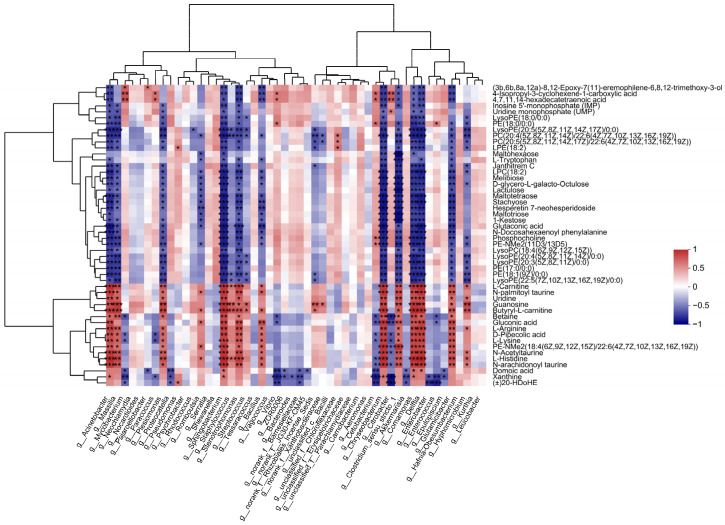
Significant correlations between intestinal bacteria at the genus level and DEMs. The correlation coefficient is represented by different colors (red, positive correlation; blue, negative correlation) with the intensity reflecting the strength as indicated in the scale. * represents significantly negative or positive correlation (* *p* < 0.05; ** *p* < 0.01; *** *p* < 0.001).

**Table 1 antioxidants-12-00198-t001:** Significantly DEMs in liver of *L. capito* after thermal stress.

Metabolites Name	Categories	VIP	FC	*p*-Value	Variation
L-Tryptophan	Amino acids	1.02	0.96	0.00	down
L-Lysine	Amino acids	1.12	1.04	0.00	up
L-Histidine	Amino acids	1.16	1.04	0.00	down
Argininosuccinic acid	Amino acids	1.18	1.07	0.00	up
L-Arginine	Amino acids	1.36	1.06	0.00	up
LysoPC(20:2(11Z,14Z))	Lipids	0.96	1.05	0.03	up
LysoPC(16:0)	Lipids	1.24	0.94	0.00	down
LysoPC(22:4(7Z,10Z,13Z,16Z))	Lipids	1.04	0.96	0.00	down
PC(22:6(4Z,7Z,10Z,13Z,16Z,19Z)/22:6(4Z,7Z,10Z,13Z,16Z,19Z))	Lipids	1.18	0.95	0.00	down
LysoPC(22:6(4Z,7Z,10Z,13Z,16Z,19Z))	Lipids	1.01	0.96	0.00	down
PC(22:5(4Z,7Z,10Z,13Z,16Z)/22:6(4Z,7Z,10Z,13Z,16Z,19Z))	Lipids	1.36	0.92	0.00	down
Arachidonic acid	Lipids	1.28	0.92	0.00	down
Gluconic acid	Carbohydrates	1.12	1.04	0.00	up
Melibiose	Carbohydrates	1.31	0.95	0.00	down
D-Tagatose	Carbohydrates	1.54	0.91	0.00	down
Uridine	Nucleosides	1.19	1.05	0.00	up
5′-CMP	Nucleosides	1.60	1.11	0.00	up
Cytidine	Nucleosides	1.36	0.92	0.00	down
Ascorbic Acid	Vitamins and Cofactors	1.39	1.13	0.00	up
D-Pantothenic acid	Vitamins	1.94	1.15	0.00	up
Uracil	Bases	1.26	1.06	0.00	up
Guanosine	Nucleic acids	1.15	1.05	0.00	up

VIP, variable importance in projection; FC, fold change calculated from the arithmetic mean values of each group. Those with VIP ≥ 1 and *p* < 0.05 were considered DEMs between two groups. A total of 210 metabolites belonging to the other classifications are not shown.

## Data Availability

The data presented in this study are available in article.

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
