# Peer review of "Effects of Thermal Stress on the Antioxidant Capacity, Blood Biochemistry, Intestinal Microbiota and Metabolomic Responses of Luciobarbus capito"

_antioxidants, 2023, doi:10.3390/antiox12010198_

Round 1

Reviewer 1 Report

General comment

In the manuscript antioxidants-2039893 entitled “Effects of thermal stress on the antioxidant capacity, immunity, intestinal microbiota and metabolomic responses of Luciobarbus capito” the Authors performed enzyme activity studies related to antioxidant and nonspecific immunity, intestinal microbiota, and untargeted LC–MS metabolomics analysis on L. capito exposed to thermal stress. Generally, the study is well structured. The discussion is well written. While I enjoyed the flow of the paper, I could not overcome the sense that there are some issues that could addressed to improve the quality of the manuscript prior to publication in Antioxidants.

Specific comments

·      I suggest to improve the introduction. In particular, the section related to oxidative stress biomarkers need to be improved (i.e., line 45). 

·      Line 46. I suggest to add a key reference at the end of this sentence “Blood plays an important role in physiological state and immune defense of fish” (i.e., https://doi.org/10.3390/ani10091466)

·      Line 69. L. capito in italics

·      Line 82. The Authors used fish to perform the experiment. Studies involving vertebrates must only be carried out after obtaining approval from the appropriate ethics committee. As a minimum, the project identification code, date of approval and name of the ethics committee or institutional review board should be stated. Research procedures must be carried out in accordance with national and institutional regulations. Statements on animal welfare should confirm that the study complied with all relevant legislation. 

·      Line 91. Please provide the size of the tanks

·      Line 155. Please add more information about the manufactures; also, QA/QC need to be added. 

·      Line 159. Please, add more information about statistical analyses. I suggest to merge the statistical analyses related to microbiome analysis (PCoA, PCA, PLS-DA etc.) 

·      Line 175. What kind of statistical test did you apply? Please, add in the caption and in the section 2.7.

Author Response

Response to Reviewer 1 Comments

We greatly appreciate the reviewers' very constructive, detailed and helpful comments and have done necessary changes according to the reviewers' advice (see our responses below on detailed comments by the reviewers).

Responses to Reviewer #1

Comment: In the manuscript antioxidants-2039893 entitled “Effects of thermal stress on the antioxidant capacity, immunity, intestinal microbiota and metabolomic responses of Luciobarbus capito. The authors performed enzyme activity studies related to antioxidant and nonspecific immunity, intestinal microbiota, and untargeted LC–MS metabolomics analysis on L. capito exposed to thermal stress. Generally, the study is well structured. The discussion is well written. While I enjoyed the flow of the paper, I could not overcome the sense that there are some issues that could addressed to improve the quality of the manuscript prior to publication in Antioxidants.

* I suggest to improve the introduction. In particular, the section related to oxidative stress biomarkers need to be improved (i.e., line 45).

Answer: Thank you for your advice. Relevant content has been improved. Line46-56.

* Line 46. I suggest to add a key reference at the end of this sentence “Blood plays an important role in physiological state and immune defense of fish” (i.e., https://doi.org/10.3390/ani10091466)

Answer: According to the suggestions of reviewer, a key reference has been added. Line 58.

* Line 69. L. capito in italics

Answer: The "L. capito" has been changed to italic. Line 79.

* Line 82. The Authors used fish to perform the experiment. Studies involving vertebrates must only be carried out after obtaining approval from the appropriate ethics committee. As a minimum, the project identification code, date of approval and name of the ethics committee or institutional review board should be stated. Research procedures must be carried out in accordance with national and institutional regulations. Statements on animal welfare should confirm that the study complied with all relevant legislation.

Answer: The study complies with all relevant laws. The project identification code is 20220420-001, the date of approval is April 30, 2022, and the name of the ethics committee is Committee for the Welfare and Ethics of Laboratory Animals of Heilongjiang River Fisheries Research Institute, CAFS. Relevant content has been added. Line 99-102. 

* Line 91. Please provide the size of the tanks

Answer: The size of the glass tank was 87.5 cm × 52.5 cm × 50.0 cm with 200 L of water. Relevant content has been added. Line 109-110.

* Line 155. Please add more information about the manufactures; also, QA/QC need to be added.

Answer: Relevant content has been added. Line 174-175.

* Line 159. Please, add more information about statistical analyses. I suggest to merge the statistical analyses related to microbiome analysis (PCoA, PCA, PLS-DA etc.)

Answer: Detailed statistical analyses has been added. Line 181-191.

* Line 175. What kind of statistical test did you apply? Please, add in the caption and in the section 2.7.

Answer: Statistical analyses were performed using SPSS 19.0. Differences between thermal exposure stress and control treatments were determined by t-test and differences were considered significant at p <0.05. Line 136-138.

Reviewer 2 Report

Overall, this is an interesting paper, with a good experimental approach, and thorough description of the methods.  My only concern is that the authors did not truly assess any immune responses or immune components, despite including “immune response” in the title and as part of the background information and discussion.  While some of the metabolites do, in fact, support immune functions, there is little support to their identification as immune metabolites. Albumin clearly has several other more prominent roles as does acid phosphatase. I think the authors should scale back the discussion of immunity in this paper in general. 

Changes in microbiota could be as much a product of the presence or absence of metabolites that the bacteria utilize as any function of shifts in immunity.

Much of the information about the role of antioxidants in resisting damage from ROS would be best suited for the introduction rather than the discussion.

P-values are not necessary in the abstract and I recommend that they be removed. Without the context of sample size and statistical tests conducted, a p-value is not meaningful, which is why they are best only presented in the results.

Line 48 “important indicators to the analyze” needs re-written

Line 69 Italicize L. capito

Line 291 Italicize Scopthalmus maximus

Line 322 Italicize Rachycentron canadum

Line 384 Italicize L. capito

Author Response

Response to Reviewer 2 Comments

We greatly appreciate the reviewers' very constructive, detailed and helpful comments and have done necessary changes according to the reviewers' advice (see our responses below on detailed comments by the reviewers).

Responses to Reviewer #2

* Overall, this is an interesting paper, with a good experimental approach, and thorough description of the methods. My only concern is that the authors did not truly assess any immune responses or immune components, despite including “immune response” in the title and as part of the background information and discussion. While some of the metabolites do, in fact, support immune functions, there is little support to their identification as immune metabolites. Albumin clearly has several other more prominent roles as does acid phosphatase. I think the authors should scale back the discussion of immunity in this paper in general.

Answer: According to the suggestions of reviewer, The expression related to immune response in the manuscript has been changed.

* Much of the information about the role of antioxidants in resisting damage from ROS would be best suited for the introduction rather than the discussion.

Answer: Some content has been transferred from the discussion to the introduction.

* P-values are not necessary in the abstract and I recommend that they be removed. Without the context of sample size and statistical tests conducted, a p-value is not meaningful, which is why they are best only presented in the results.

Answer: p-value has been deleted in the abstract.

Line 48 “important indicators to the analyze” needs re-written

Answer: The sentence has been rewritten. Line 59.

* Line 69 Italicize L. capito

Answer: The "L. capito" has been changed to italic. Line 82.

* Line 291 Italicize Scopthalmus maximus

Answer: The "Scopthalmus maximus" has been changed to italic.Line 324.

* Line 322 Italicize Rachycentron canadum

Answer: The "Rachycentron canadum" has been changed to italic. Line 359.

* Line 384 Italicize L. capito

Answer: The "L. capito" has been changed to italic.Line 442.

Reviewer 3 Report

Effects of thermal stress on the antioxidant capacity, immunity, intestinal microbiota and metabolomic responses of Luciobarbus capito

While the authors use an interesting approach to elaborate impact of short time temperature elevations in L capito, with clear results, the biological interpretation and meaning are not convincing. A lot of basic knowledge is missing which generally ends up with a poor discussion. The experiment is also poorly described, and the language is imprecise, which often leads to misunderstandings.

L14 unprecise statement

L16  intestinal microbiota and hepatopancreas LC–MS metabolomics (?)

L24 should indicate mechanisms

L45 Please be precise –“ induce immune responses in fish”.

L47 Blood index is not defined - Hematology? Clinical analyses?

L50 Hepatopancreas or liver?

L95 No start sampling ?

L104 Duplicate means replicate?

L108 Where are data on body weight, hepatopancreas weight and HP relative to body weight? Could indicate tissue edema

L113 Analyses for serum?

L115 analyzer is not specified.

L158 No statistical information on the serum (SD ++ ?) and metabolome analyses (PCA ++ ?)

L171 Imprecise legend (tissue? Statistics?)

L235 Imprecise fig legend

L279-280: What are the indications of oxidative damage?

L293: elevated s-glucose indicate a secondary stress response; may be elevated since the fish starved through the experiment; may be elevated due to dehydration. Please discuss.

L301. The elevated enzymes may indicate increased deamination for easy accessible energy.  Or increase as an osmotic response to stress (like alb and ACP). Please discuss.

L 308. This is not likely. Osmotic difference?

L316 – no discussion of the effect of temperature on bacterial growth, especially since the fish was not fed

L342 – Circular argumentation on why amino acid metabolism is affected by temperature. Free amino acids (histidine, arginine and lysine) have many biological functions beyond serving as substrates for protein synthesis and energy. Please elaborate this against updated literature.

L358 – disruptive effects on lipid metabolism or homeoviscous adaptation of membranes according to temperature?? Poor discussion.

Author Response

Response to Reviewer 3 Comments

We greatly appreciate the reviewers' very constructive, detailed and helpful comments and have done necessary changes according to the reviewers' advice (see our responses below on detailed comments by the reviewers).

Responses to Reviewer #3

* L14 unprecise statement

Answer: This sentence has been modified. Line 14.

* L16 intestinal microbiota and hepatopancreas LC–MS metabolomics (?)

Answer: This sentence has been revised according to the suggestions of the reviewer. Line 17.

* L24 should indicate mechanisms

Answer: The sentence has been rewritten according to the suggestions of the reviewer. Line 25-27.

* L45 Please be precise –“ induce immune responses in fish”.

Answer: This sentence is redundant and has been deleted.

* L47 Blood index is not defined - Hematology? Clinical analyses?

Answer: The sentence has been rewritten.

* L50 Hepatopancreas or liver?

Answer: The correct expression is "liver" It has also been modified elsewhere in the manuscript.

* L95 No start sampling?

Answer: The sampling time was 96 hours after the stress. Detailed description is given in Section 2.3.

* L104 Duplicate means replicate?

Answer: Yes. To avoid misunderstanding, "duplicate" has been replaced by "biologically repeated".

* L108 Where are data on body weight, hepatopancreas weight and HP relative to body weight? Could indicate tissue edema

Answer: 0.20g of liver tissue were weighed and then homogenized with isotonic saline, which can't cause tissue edema. This sentence has been modified. Line 127.

* L113 Analyses for serum?

Answer: I'm sorry for the misunderstanding caused by the author's negligence. We measured the antioxidant enzymes in liver and biochemical indicators in serum. Relevant contents have been supplemented in the manuscript.

* L115 analyzer is not specified.

Answer: The model of the analyzer has been added to the manuscript. Line 135.

* L158 No statistical information on the serum (SD ++ ?) and metabolome analyses (PCA ++ ?)

Answer: The missing content has been added to the manuscript.

* L171 Imprecise legend (tissue? Statistics?)

Answer: The legend has been modified.

* L279-280: What are the indications of oxidative damage?

Answer: The sentence has been modified. The significant increase of MDA content and decrease of antioxidant enzyme activity were the indications of oxidative damage.

* L293: elevated s-glucose indicate a secondary stress response; may be elevated since the fish starved through the experiment; may be elevated due to dehydration. Please discuss.

Answer: This part has been modified. Please read the manuscript.

* L301. The elevated enzymes may indicate increased deamination for easy accessible energy. Or increase as an osmotic response to stress (like alb and ACP). Please discuss.

Answer: As the reviewer said, fish can increase deamination for easy accessible energy. However, this process mainly occurs in the liver, with the rise of ALT and AST. Under normal physiological metabolic conditions, ALT and AST mainly exist in the liver, while serum ALT and AST levels are low and relatively stable. Unfortunately, the ALT and AST levels in the liver were not measured in this study. The references we cited can prove this point.

L316 – no discussion of the effect of temperature on bacterial growth, especially since the fish was not fed

Answer: This part has been modified. Please read the manuscript.

L342 – Circular argumentation on why amino acid metabolism is affected by temperature. Free amino acids (histidine, arginine and lysine) have many biological functions beyond serving as substrates for protein synthesis and energy. Please elaborate this against updated literature.

Answer: This part has been modified. Please read the manuscript.

L358 – disruptive effects on lipid metabolism or homeoviscous adaptation of membranes according to temperature?? Poor discussion.

Answer: This part has been modified. Please read the manuscript.

Round 2

Reviewer 1 Report

I noticed that important improvements and clarifications were provided by the Authors. Furthermore, the Authors have satisfactorily responded to all my comments and suggestions. The MS is now ready for publication in Antioxidants.